# Mechanical and Thermal Properties of Synthetic Polypropylene Fiber–Reinforced Renewable Oil Palm Shell Lightweight Concrete

**DOI:** 10.3390/ma14092337

**Published:** 2021-04-30

**Authors:** Leong Tatt Loh, Ming Kun Yew, Ming Chian Yew, Jing Han Beh, Foo Wei Lee, Siong Kang Lim, Kok Zee Kwong

**Affiliations:** 1Department of Civil Engineering, Lee Kong Chian Faculty of Engineering and Science, Universiti Tunku Abdul Rahman, Cheras, Kajang 43000, Malaysia; dylantatt06@gmail.com (L.T.L.); Leefw@utar.edu.my (F.W.L.); sklim@utar.edu.my (S.K.L.); kwongkz@utar.edu.my (K.Z.K.); 2Department of Mechanical and Material Engineering, Lee Kong Chian Faculty of Engineering and Science, Universiti Tunku Abdul Rahman, Cheras, Kajang 43000, Malaysia; yewmc@utar.edu.my; 3Department of Architecture and Sustainable Design, Lee Kong Chian Faculty of Engineering and Science, Universiti Tunku Abdul Rahman, Cheras, Kajang 43000, Malaysia; behjh@utar.edu.my

**Keywords:** oil palm shell, lightweight concrete, thermal conductivity, polypropylene fiber, mechanical properties

## Abstract

Oil palm shell (OPS) is an agricultural solid waste from the extraction process of palm oil. All these wastes from industry pose serious disposal issues for the environment. This research aims to promote the replacement of conventional coarse aggregates with eco-friendly OPS aggregate which offers several advantages, such as being lightweight, renewable, and domestically available. This paper evaluates the mechanical and thermal performances of renewable OPS lightweight concrete (LWC) reinforced with various type of synthetic polypropylene (SPP) fibers. Monofilament polypropylene (MPS) and barchip polypropylene straight (BPS) were added to concrete at different volume fractions (singly and hybrid) of 0%, 0.1%, 0.3% and 0.4%. All specimens were mixed by using a new mixing method with a time saving of up to 14.3% compared to conventional mixing methods. The effects of SPP fibers on the mechanical properties were investigated by compressive strength, splitting tensile strength and residual strength. The strength of the oil palm shell lightweight concrete hybrid 0.4% (OPSLWC–HYB–0.4%) mixture achieved the highest compressive strength of 29 MPa at 28 days. The inclusion of 0.3% of BPS showed a positive outcome with the lowest thermal conductivity value at 0.55 W/m °C. Therefore, the results revealed that incorporation of BPS fiber enhanced the performance of thermal conductivity tests as compared to inclusion of MPS fiber. Hence, renewable OPS LWC was proven to be a highly recommended environmentally friendly aggregate as an alternative solution to replace natural aggregates used in the concrete industry.

## 1. Introduction

The world’s population is projected to reach approximately nine billion in the next 30 years [1]. The blooming of population will uplift the construction industry due to the escalating demand for shelters and infrastructure. Among the components of concrete, 55−80% of concrete volume is occupied by aggregate. Hence, the production of concrete has consumed 8 to 12 billion tons of natural aggregates per annum after 2010 [2]. The scarcity of natural resources has become a global concern today. Therefore, the search for alternative renewable sources to replace conventional aggregate has been initiated. One of the potential substituents is the agricultural solid waste from palm oil manufacturers, oil palm shell (OPS). The disposal of solid waste from palm oil industry has imposed burden to the environment. Thus, the utilization of OPS as coarse aggregate in concrete can be one of the best possible solutions to relieve the waste disposal problem and the depletion of natural aggregate [3].

The oil palm trees are grown abundantly in tropical regions, such as Indoensia, Malaysia, Thailand, Colombia, Ecuador and Papua New Guinea [4]. Malaysia is the second largest exporter of palm oil, which accounts for 36.75% of the global palm oil trade [5]. At the same time, the biomass waste produced from the manufacturing process of palm oil accounts for 85.5% of the total production of biomass in Malaysia [6]. Malaysia Palm Oil Board stated that approximately 4.18 million tons of OPS was produced in 2016. Research on replacing conventional coarse aggregate by OPS as lightweight aggregate (LWA) has been pioneered in Malaysia since 1984 [7]. It has been reported that lightweight aggregate concrete (LWAC) has a water to cement ratio not more than 0.45 and oven-dry density less than 2000 kg/m^3^ [8]. The advantages of using OPS as LWA are domestic availability, renewable and low cost. Application of OPS as aggregate substituent has been demonstrated in the construction of a small foot bridge in 2001 and a model low-cost house in 2003 by using “OPS hollow blocks” as walls and “OPS concrete” as footings, lintels and beams in Sabah, Malaysia [9]. 

It has been reported that mechanical properties such as compressive strength, splitting tensile strength, flexural strength and modulus of elasticity of oil palm shell lightweight concrete (OPSLWC) are lower than other lightweight aggregate concrete (LWAC) [10,11]. Therefore, the incorporation of discontinuous fibres (polyvinyl alcohol, polypropylene, nylon and steel) into OPSLWC has been proposed to improve concrete mechanical performance [12,13,14,15,16,17,18]. According to thr previous studies, the inclusion of highly dense metallic fibers effectively enhanced the mechanical properties, but it also significantly deteriorated the workability and caused an increment in density, which does not favor the development of LWC [19,20]. Therefore, synthetic polypropylene fibers (SPP) were chosen for this research. It has been reported that the inclusion of polypropylene fibers showed enhancement in mechanical and thermal properties [21,22]. However, very limited work on the role of nanomaterials as supplementary cementitious materials, such as metakaolin, fly ash or natural pozzolans, are used [23] instead of cement into oil palm shell lightweight concrete. Therefore, the potential of oil palm shell ash and chicken eggshell powder with nano particle size will be investigated for the future works.

This paper evaluates the effects of two different SPP fibers on the mechanical and thermal properties of OPSLWC. The fibers were added singly and hybrid in different volume fractions (0.1%, 0.3% and 0.1% + 0.3%). This study focuses on investigating the mechanical characteristics, namely the compressive strength, splitting tensile strength, flexural strength and residual strength (compression and tension), and the thermal conductivity of OPSLWC.

## 2. Materials and Methods

### 2.1. Materials 

#### 2.1.1. Cement

Type 1 Ordinary Portland Cement (OPC) manufactured by Tasek Corporation Berhad, in compliance with MS EN 197–1:2014, was used for all the mixing in this research. The technical specifications of the OPC used are summarized in Table 1. 

#### 2.1.2. Mixing Water and Superplasticizers (SP)

Potable water from a municipal tap supplied by the local water supplier was utilized for mixing and curing. The quality of water was monitored to ensure it was free from any contaminants or impurities, as these contaminants/impurities can adversely affect the hydration of cement and the properties of concrete. The specific gravity of the mixing water was taken at 1 g/cm^3^. High performance water reducer, Master Glenium^®^ SKY 8808 manufactured by BASF was added to all the mixes at a constant amount of 2.0% of the cement weight to enhance the workability of fresh concrete.

#### 2.1.3. Aggregates (Sand and OPS) 

In this research, conventional coarse aggregates for all the mixes were fully substituted by crushed older OPS, which had been thrown out for more than 180 days. The fiber content in the surface of old OPS is less than 2%, which offers a stronger bonding between the cement and OPS grains resulting in a better strength performance [12,24]. Before mixing, the crushed OPS were sieved through a 10-mm sieve to remove OPS aggregates with sizes greater than 10 mm and completely immersed in water for 24 h. Aggregates were air dried for 3 h at room temperature after removing from the water to attain the saturated surface dry (SSD) condition. This measure was taken to prevent extra water generated on the surface of OPS aggregates, which may influence the water–cement (w/c) ratio and the strength properties of the concrete. The physical properties of the coarse aggregates are listed in Table 2. Domestic mining sand was used as fine aggregate in this study. The specific gravity, fineness modulus, water absorption and maximum grain size were 2.63 g/cm^3^, 2.75, 0.95% and 4.75 mm, respectively. Sieve analysis was performed according to ASTM C 136–01 in order to obtain the grading of fine aggregate used in this study. Average distribution of sand was obtained by performing two sieve analyses, using a similar source, as shown in Figure 1. The 4.75 mm aggregates were kept in a plastic container with a cover after being dried under the sun. This was to prevent the 4.75 mm aggregates from being exposed to water vapor in the air.

#### 2.1.4. Synthetic Polypropylene Fibers

Synthetic polypropylene (SPP) fiber is derived from monomeric hydrocarbon C_3_H_6_, which has a low specific gravity of 0.91 g/cm^3^. The good mechanical properties of SPP fibers offer a tensile strength of 600 MPa and Young’s modulus of 4.11 GPa. The mode of polymerization gives polypropylene a sterically regular atomic arrangement that results in its chemical inertness. Hence, it can be added into a highly alkaline matrix without any adverse effects on its properties. In addition, SPP is a good thermal insulator, and has a low thermal conductivity of 0.1–0.22 W/mK. In this research, two different types of SPP fibers, monofilament polypropylene straight (MPS) and barchip polypropylene straight (BPS), were added to the OPSLWC. The properties of these polypropylene fibers are tabulated in Table 3. 

### 2.2. Design of the Mix

Four different mixes were prepared as shown in Table 4. The proportions of cement, water, fine aggregates, OPS and SP were kept constant in all mixes, which were 515 kg/m^3^, 170 kg/m^3^, 1000 kg/m^3^, 290 kg/m^3^ and 2.0% of the binder weight, respectively. Different types and volume fractions of fibers were added to different mixes. Each mix was given a mix code for the ease of identification. 

### 2.3. Experimental Procedures

The preparation procedures of all the mixes are described in this section. The trial mixing method was conducted to make a comparison between conventional mixing method (CMM) and new mixing method (NMM). From the results, NMM, in terms of total mixing time, could save up to 60 s compared to CMM. Therefore, a new mixing method was used to produce all the mixes [25], as shown in Figure 2**.** This mixing method was developed by modifying the mix proportion of self-compacting concrete (SCC), which comprises a liquid phase and solid phase with a total mixing time of 360 s [26]. The fibers with different percentage of volume fractions (0.1, 0.3 and 0.4%) were then distributed and mixed for 180 s in the mix. Slump tests and Vebe tests were carried out in accordance with BS EN: 12350–Part 2: 2009 and BS EN: 12350–Part 3: 2009, respectively, to determine the workability of all mixes. Oil was applied on all the surfaces of the molds before casting. The molds filled with slump were vibrated on a shaking table to ensure uniformity of the mix. The concrete samples were demolded after 24 h from the time of placing. All the demolded samples were fully immersed in water at room temperature in a curing tank until they reached the desired testing age.

A 3000 kN capacity compression test machine was used and was manufactured by Unit Test Scientific Sdn. Bhd. It was set to a constant loading rate of 3.0 kN/s in compliance with BS EN 12390–Part 3: 2009. The same machine was used for the splitting tension tests with a loading rate set to 1.5 kN/s in accordance with BS EN 12390–Part 6: 2009. For each mixture sample, cubes with dimensions of 100 mm × 100 mm × 100 mm were cast to test the compressive strength for 1, 7, 28, 90 and 180 days. Three cylinders with dimensions of 100 mm × 200 mm were cast to test the splitting tensile strength of the mixture samples at 7, 28 and 90 days. Furthermore, the residual strengths of the cube compressive and splitting tensile strengths were determined by loading the specimens for a second and third time and set at a 10% failure rate.

The thermal conductivity was determined by using a guarded hot plate and a heat flow meter methods according to BS EN 12664: 2001, as shown in Figure 3.

## 3. Result and Discussion

### 3.1. Workability

The workability of concrete plays an important role in determining the homogeneity at which it can be blended, placed, compacted, and finished. In this research, the workability of mixes was measured through a slump test and Vebe test. The results collected from both tests are tabulated in Table 5.

The slump value can be correlated to the Vebe time, as shown in Figure 4, with an R^2^ value of 0.845. Hence, Equation (1) was proposed to predict the slump value with the given Vebe time.
(1)$$S=19133{\left(V\right)}^{-2.101}$$
where *S* denotes the slump value in cm and V indicates the Vebe time in s. 

### 3.2. Density

Demolded density (DD) and oven-dried density (ODD) were measured for all mixes, as shown in Table 6. DD was calculated using the weight of the specimens, measured after demolding; while ODD was calculated with the weight of the 28-day specimens, measured after being oven dried at 105 °C for 24 h. All specimens in this study were found to have DD and ODD in the range of 2047.4–2095.0 kg/m^3^ and 1936.7–1976.5 kg/m^3^, respectively. The outcome fulfilled the objective of obtaining OPSLWC with an ODD less than 2000 kg/m^3^. The samples also met the requirements for structural applications as structural lightweight concrete (SLWC), defined as concrete with an ODD no greater than 2000 kg/m^3^ [27].

### 3.3. Compressive Strength

The compressive strength of each mix at 1, 7, 28, 56, 90 and 180 days is shown in Table 7. The 28-day compressive strength of all mixes was in the range of 25 to 30 MPa, which met the requirement for structural lightweight concretes (SLWC) [27]. The inclusion of PP fibers enhanced the compressive strength by 4.8–17.4% at 28 days and 11.9–17% at 180 days.

When a compressive load is applied to a specimen, lateral tension is produced randomly throughout the sample. Minor cracks will start to develop due to volumetric changes in the heterogenous mixture of concrete and can lead to hairline cracks. Minor cracking will continue to occur and eventually major cracks will form when the advancing minor cracks meet each other. The cracking will take place in the direction parallel to the applied load. The presence of PP fibers blocks the propagation of the crack and acts as a resistance to lateral tension. As a result, additional compressive loading can be withstood, and thus the capacity of concrete to withstand compression was improved. This showed the attributes of fibers in arresting cracks and bridging concrete more effectively [14,28,29].

Figure 5 illustrates the growth of compressive strength with respect to the age of the mixes. It can be noticed that the compressive strength of all mixes increased with age. From 28 days to 90 days, the compressive strength of all mixes gradually improved. Compressive strength of all mixes increased from 7–14.7% at 180 days. This phenomenon could be due to the new mixing method (NMM). The expedited reaction of SP and cement could result a more thorough dispersion of cement particles to ensure homogeneous distribution and improve the stability of the cement hydration reaction.

### 3.4. Residual Compressive Strength

The residual compressive strength (RCS) is a streamlined method to evaluate the effects of fibers on the toughness of concrete. This test was conducted by further loading the specimens a second and third time at 28 days. RCS shows a clearer comparison and the positive effects of the different types of PP fibers on the post-cracking of fiber-reinforced oil palm shell lightweight concrete (FROPSLWC). The values of the first loading compressive strength (FLCS), second loading compressive strength (SLCS) and third loading compressive strength (TLCS) are tabulated in Table 8. 

Without the incorporation of fibers, OPSLWC–CTR–0% failed at 64.8% of FLCS and 40.9% of FLCS for the second and third loadings, respectively. The inclusion of fibers showed a remarkable improvement of 17.8% to 24.2% for SLCS and 33.5% to 42.4% for TLCS. Comparing OPSLWC–BPS–0.3% and OPSLWC–MPS–0.1%, the former produced higher SLCS and TLCS of 87% and 74.7%, respectively. The latter retained SLCS of 82.6% and TLCS of 74.4%. This could be due to the higher volume fraction and different geometry of BPS [30]. The mix with hybrid fibers, OPSLWC–HYB–0.4%, had the highest RCS values of 89% and 83.3% for SLCS and TLCS, respectively. This could be due to the synergy effect between the two different fibers. From a geometry point of view, MPS was smaller and more effective in arresting minor cracks, while BPS was bigger and stiffer and more effective in arresting major cracks.

The failure pattern of OPSLWC with fibers and without fibers is illustrated in Figure 6. The cube specimen without fibers underwent semi-explosive failure; while the cube specimen with fibers underwent non-explosive failure [14,30]. From the Figure 7, the linking bridge between fibers and cement matrices can prevent the spalling caused by the lateral tension induced by compressive loading [20]. The holding effect of fibers contributed to the improvement of RCS. 

### 3.5. Splitting Tensile Strength

Splitting tensile strength was tested at 7 days, 28 days and 90 days for all mixes. The splitting tensile strength increased with the ages of concrete, as shown in Figure 8. Incorporation of hybrid MPS and BPS with a total volume fractions of 0.4% enhanced splitting tensile strength by 3.4% at 28 days and 8.7% at 90 days as compared to the control mix. Addition of 0.3% of BPS upgraded splitting tensile strength by 1.4% at 28 days and 7.8% at 90 days. Furthermore, inclusion of 0.1% of MPS improved 0.5% splitting tensile strength at 28 days and 1.4% at 90 days, respectively. 

Splitting tensile strength of OPSLWC was relatively lower than that of concrete with normal aggregate due to the weak bonding between the OPS and cement matrix [12]. Therefore, polypropylene fibers were added to enhance the tensile strength of OPSLWC. In Figure 7, the trend of the development of splitting tensile strength is clearly presented, which shows that strength increased with the addition of volume fraction of fibers. Upon loading, the cylindrical specimen without fibers tended to split in half due to the tensile stress induced by the load. The presence of fibers prevented the cylindrical specimen from splitting. The polypropylene fibers with a high tensile strength stretched upon splitting and achieved a strong fiber–matrix interfacial adhesion to hold the cement. As a consequence, a higher splitting load was required to break the specimen in half. The stretching of fibers upon splitting is illustrated in Figure 9.

### 3.6. Residual Splitting Tensile Strength

Residual splitting tensile strength (RSTS) is another method to investigate the effects of fibers on the integrity of concrete. Without the presence of fibers, OPSLWC–CTR–0% split in half after the first loading, as shown in Figure 10. However, the other fiber-reinforced mixes (OPSLWC–MPS–0.1%, OPSLWC–BPS–0.3% and OPSLWC–HYB–0.4%) were not split in half after the first loading. Thus, OPS fiber-reinforced lightweight concretes proceeded to the second and third loadings for the determination of residual strength at 28 days. First loading splitting tensile strength (FLSTS), second loading splitting tensile strength (SLSTS) and third loading splitting tensile strength (TLSTS) of all mixes at 28 days are shown in Table 9. 

From Table 9, OPSLWC–MPS–0.1% retained 50.7% of its strength at the second loading and 29.2% at third loading. Higher strength was retained for OPSLWC–BPS–0.3% (62.6% at second loading and 48.3% at third loading) as the geometry of BPS provided a better bridging effect compared to MPS. It can be seen that OPSLWC–HYB–0.4% had the highest residual strength of 80.9% at the second loading and 66.5% at the third loading. This result could be attributed to the combined effect of two different types of fibers on OPSLWC. Although the improvement in FLSTS was not incredibly significant, the enhancement in SLSTS and TLSTS was very pronounced with an increase in retained strength of 18.3–30.2% in the second loading and 18.2%–37.3% in the third loading, as compared to the single type of OPS fiber-reinforced lightweight concrete. 

The failure mode of the specimen differed from the control mix in fiber-reinforcement, as shown in Figure 11. Apart from the splitting failure, shear failure occurred in the specimen with fiber-reinforced lightweight concrete. This could be attributed to the enhancement of RSTS incorporating fibers. Based on previous studies, this could be due to the bridging effect of the fibers being strong enough to hold the specimen from splitting completely [29,31]. The fibers enabled more stress transfer among the matrix until the specimen eventually failed due to shear. The fibers provided extra capacity for concrete to withstand more loads, and hence extended the shear failure. 

### 3.7. Flexural Strength

Figure 12 shows that the flexural strength increased with the volume fractions of fibers and their age. The increase ranged from 15% to 15.9% with the inclusion of fibers. It marked a significant improvement of 24.3% to 38.8%. The beneficial effect of the addition of fibers could be justified by the cracking control properties of PP fibers. As the prism specimen is under bending uniaxial loading, tensile stress will be induced at the bottom of the prism and cause the cracking to start at the micro scale. The micro cracks propagate rapidly in parallel directions to the applied load. Without the presence of fibers, the prism splits into pieces immediately. Concrete is prone to sudden failure when subjected to bending. Therefore, the presence of fibers acted as an inhibitor to the propagation of cracks. The integration of the cement matrices by the incorporated fibers reduced the propagation of cracks. Upon continuous loading, the development of cracks was initiated at the micro scale. As these micro cracks propagated and joined together to form macro cracks, the fibers stretched themselves to accommodate the crack face separation. This induced the formation of a linking bridge to hold the openings by displacing minor cracks. The stretching of fibers allowed stress distribution and an additional energy absorbing mechanism. Eventually, failure was delayed with a greater tolerance to deformation. Thus, the load capacity increased and the MOR was improved.

### 3.8. Thermal Conductivity

The thermal conductivity increased when curing age increased. The thermal conductivity of the 28-day mix was higher than the thermal conductivity of a similar 7-day mix. Voids were present in the mix at 7 days due to the cement being in a partially hydrated state [32]. Further hydration of cement in the mix at 28 days caused calcium silica hydrate to fill up most of the voids in the mix. Therefore, this resulted in higher thermal conductivity as the voids were reduced. From the results, the increase in thermal conductivity was due to the increase in the solid phases, which have shown higher thermal conductivity. 

As shown in Figure 13, the 28-day thermal conductivity of OPSLWC–MPS–0.1% was highest at 0.81 W/m °C. Although the incorporated fibers were low in terms of thermal conductivity, due to the micro geometry and the small amount of MPS fibers, the addition of fibers could fill the voids of the binder matrix and enhance the thermal conductivity [21,32]. For OPSLWC–BPS–0.3%, the 28-day thermal conductivity was lower than the control mix. The presence of a lower thermal conductivity of BPS fibers that were randomly distributed in the concrete lowered the overall thermal conductivity of the concrete [33]. However, the thermal conductivity of OPSLWC–HYB–0.4% increased to 0.68 W/m °C. In the presence of both MPS and BPS fibers, MPS filled the voids and led to a rise in thermal conductivity. On the contrary, BPS contributed to lowering the overall thermal transferability. Hence, the effect of geometrical properties of the fibers was proven. The thermal transferability of OPSLWC–HYB–0.4% lay between that of OPSLWC–MPS–0.1% and OPSLWC–BPS–0.3%. In a nutshell, the incorporation of BPS fiber was the most effective in lowering thermal conductivity compared to the other mixes.

## 4. Conclusions

Based on the results acquired from the properties of synthetic polypropylene fiber-reinforced renewable OPSLWC, the following conclusions can be made:The inclusion of synthetic polypropylene fibers deteriorated the workability. The slump value reduced from 165 mm to 96 mm and the Vebe time increased from 9.46 s to 12.26 s.Incorporation of synthetic polypropylene fibers in OPSLWC offered a positive effect on the mechanical properties. The bridging effect of fibers contributed to arresting the propagation of cracks, enabling stress transfer, providing an extra energy-absorbing mechanism, and was thus able to accommodate larger deformations. The presence of fibers also modified the failure mode of OPSLWC under compression and splitting tensions.The positive effect of fibers on the compressive strength was more pronounced in OPSLWC–HYB–0.4% with the highest volume fraction. Enhancements of 4.8% to 17.4% and 11.9% to 17% were observed in 28 days and 180 days, respectively.The inclusion of fibers improved the splitting tensile strength in the range of 0.5–3.4% at 28 days and 1.4–8.7% at 90 days.The hybrid synthetic polypropylene fiber showed the most pronounced effect in enhancing residual strength for both compressive strength and splitting tensile strength.The effect of fibers on thermal conductivity varied depending on the type of fiber added. Monofilament polypropylene straight (MPS) increased thermal conductivity, while barchip polypropylene straight (BPS) lowered the thermal conductivity.The flexural strength of OPSLWC was improved by 24.3% to 38.8% with the reinforcement of fibers.

It can be concluded that the findings of synthetic polypropylene fiber-reinforced renewable OPS lightweight concrete demonstrated a great potential application in building construction.

## Figures and Tables

**Figure 1 materials-14-02337-f001:**
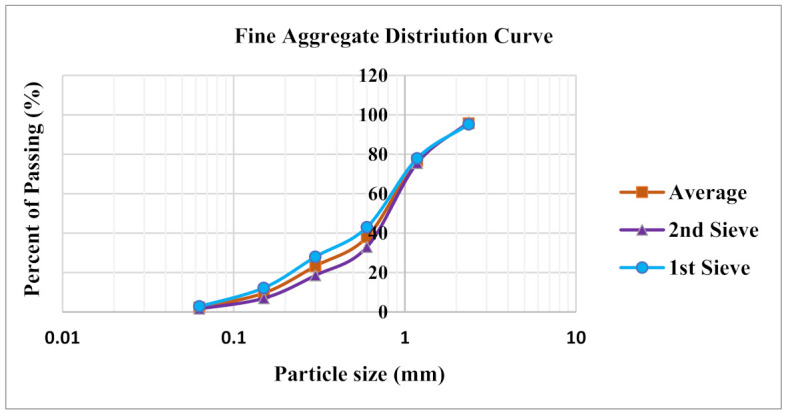
Fine aggregate distribution curve.

**Figure 2 materials-14-02337-f002:**
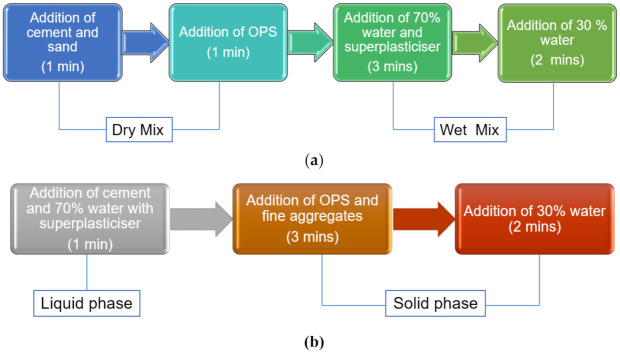
Flowchart of the (**a**) conventional mixing method (CMM) and (**b**) the new mixing method (NMM).

**Figure 3 materials-14-02337-f003:**
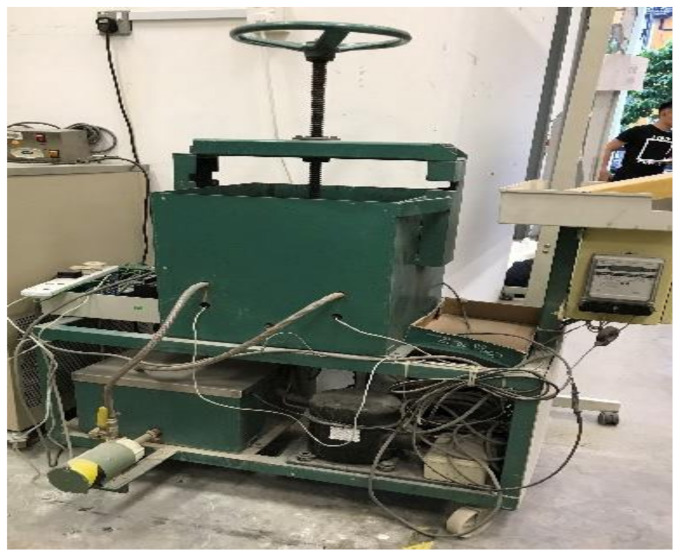
Test equipment for thermal conductivity.

**Figure 4 materials-14-02337-f004:**
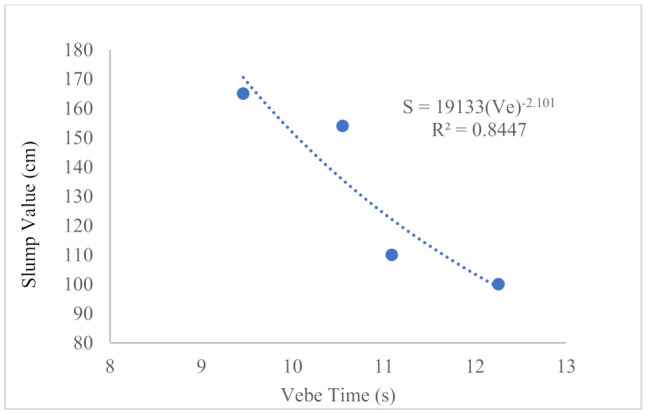
Relationship between slump value (cm) and Vebe time (s).

**Figure 5 materials-14-02337-f005:**
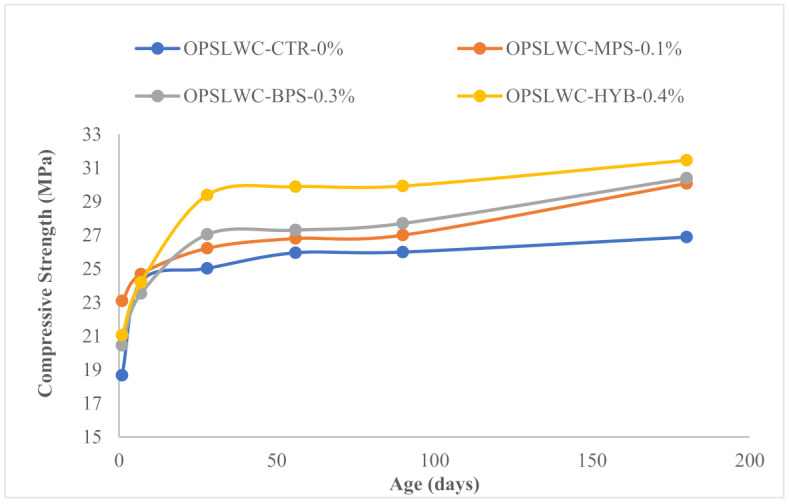
Development of compressive strength at different ages.

**Figure 6 materials-14-02337-f006:**
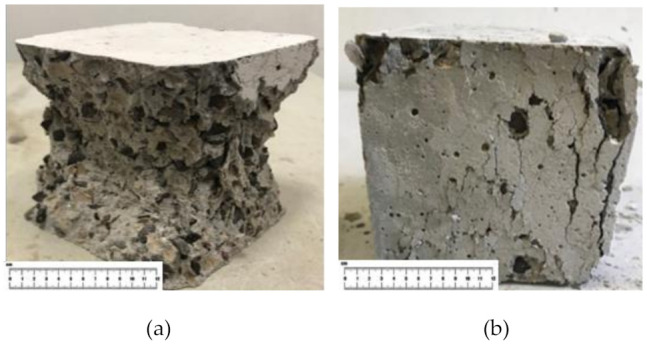
Failure pattern of OPSLWC (**a**) without fibers and (**b**) with fibers after the third loading.

**Figure 7 materials-14-02337-f007:**
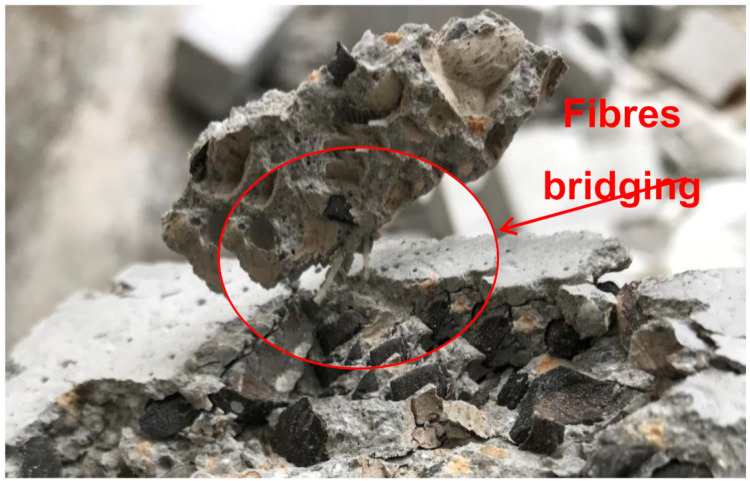
Link bridge connection between fibers and cement matrices.

**Figure 8 materials-14-02337-f008:**
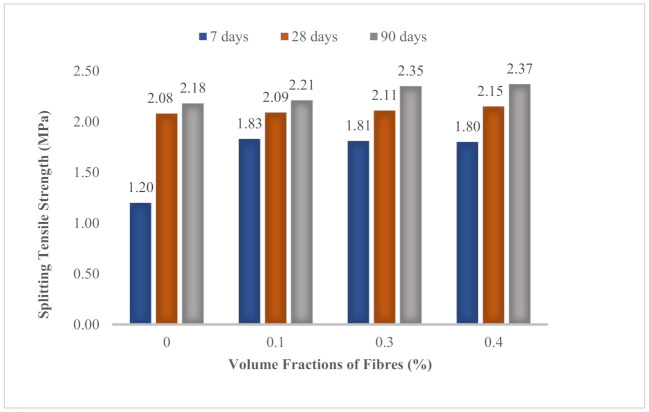
Development of splitting tensile strength of all mixes.

**Figure 9 materials-14-02337-f009:**
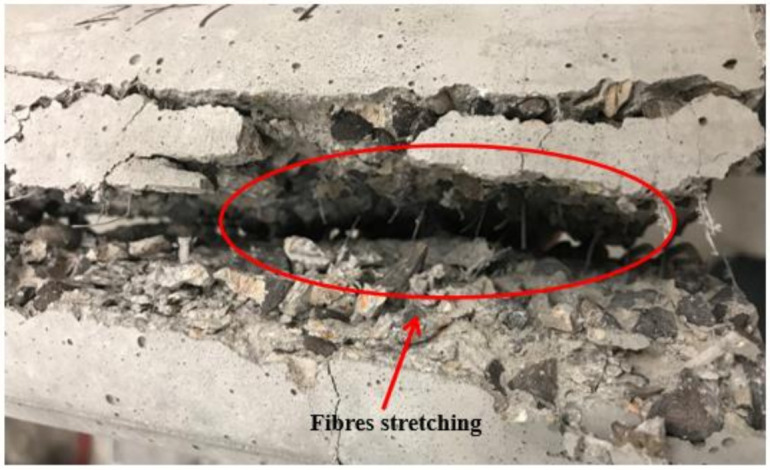
Stretching of reinforced fibers upon splitting.

**Figure 10 materials-14-02337-f010:**
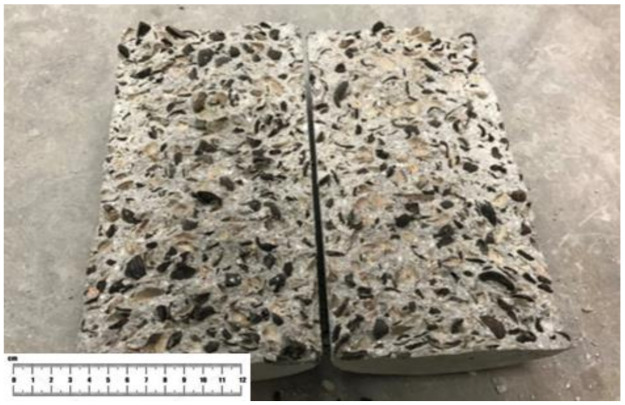
Failure mode of OPSLWC–CTR–0% after first loading.

**Figure 11 materials-14-02337-f011:**
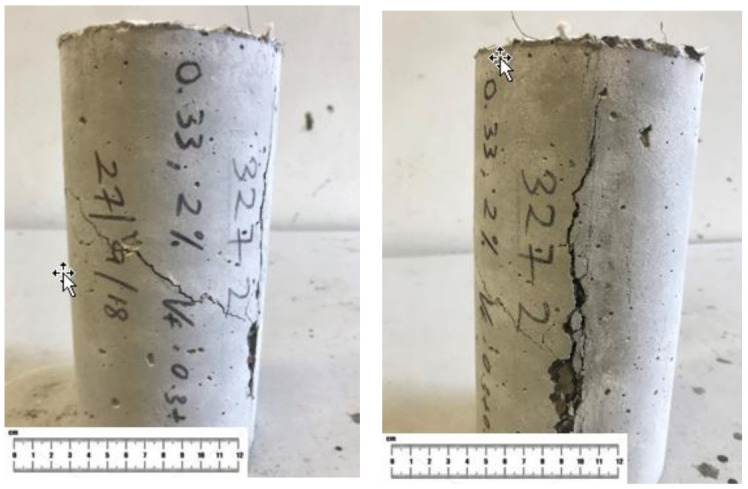
Failure mode of OPSLWC–HYB–0.4% after the third loading.

**Figure 12 materials-14-02337-f012:**
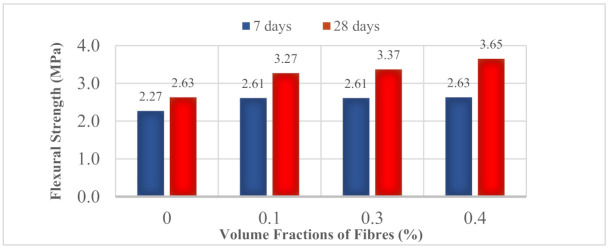
Comparison of 7- and 28-day flexural strength of different mixes.

**Figure 13 materials-14-02337-f013:**
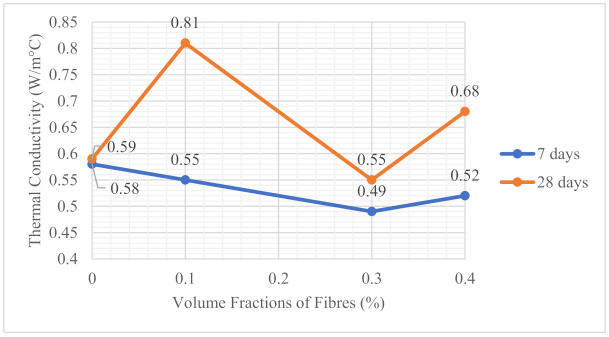
Thermal conductivity of OPSLWC incorporated with various percentages of fibers at 7 days and 28 days.

**Table 1 materials-14-02337-t001:** Specifications of OPC.

Physical Specifications			Chemical Composition (%)					
LOI	Specific Gravity	Blaine’s specific surface area (cm^2^/g)	SiO_2_	CaO	Fe_2_O_3_	MgO	Al_2_O_3_	SO_3_
0.64	3.14	3510	21.28	64.64	3.36	2.06	5.60	2.14

**Table 2 materials-14-02337-t002:** Physical properties of OPS.

Physical Property	OPS	Unit
Maximum size	10	mm
Specific gravity (SSD state)	1.30	g/cm^3^
Aggregate impact value	2.33	%
Compacted bulk density	632	kg/m^3^
Water absorption	2.34	%

**Table 3 materials-14-02337-t003:** Specifications of polypropylene fibers.

Fibres	Fibre Type	Length (mm)	Diameter (mm)	Aspect Ratio
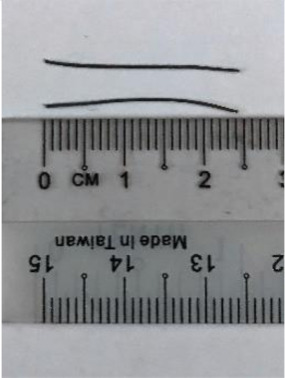	Monofilament polypropylene straight (MPS)	25	0.5	50
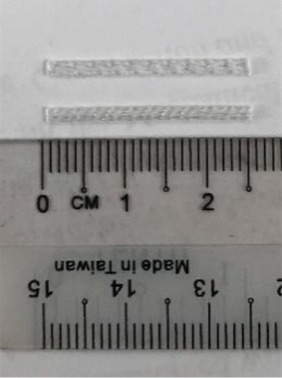	Barchip polypropylene straight(BPS)	25	1.5	17

**Table 4 materials-14-02337-t004:** Mix codes.

Mix Codes.	Types of Fibres Added	Volume Fractions (%)
OPSLWC ^1^–CTR ^2–^0% ^3^	–	0
OPSLWC–MPS–0.1%	MPS	0.1
OPSLWC–BPS–0.3%	BPS	0.3
OPSLWC–HYB–0.4%	MPS + BPS	0.1 + 0.3

Note: ^1^ “OPSLWC” denotes oil palm shell lightweight concrete. ^2^ Types of fibers added into the mix; “CTR” refers to “control”, which means no fibers added; “HYB” defines as “hybrid”, which means both types of fibers are added. ^3^ Volume fractions of fibers added.

**Table 5 materials-14-02337-t005:** Workability of all mixes.

Mix Code	Workability	
	Slump Value (cm)	Vebe Time (s)
OPSLWC–CTR–0%	165	9.46
OPSLWC–MPS–0.1%	154	10.55
OPSLWC–BPS–0.3%	110	11.09
OPSLWC–HYB–0.4%	96	12.26

**Table 6 materials-14-02337-t006:** Densities of all mixes.

Mix Code	DD (kg/m^3^)	ODD (kg/m^3^)
OPSLWC–CTR–0%	2095.0	1976.5
OPSLWC–MPS–0.1%	2088.6	1963.8
OPSLWC–BPS–0.3%	2062.8	1952.4
OPSLWC–HYB–0.4%	2047.4	1936.7

**Table 7 materials-14-02337-t007:** Compressive strength of each mix at different ages.

Mix Code	Compressive Strength (MPa)					
	1 day	7 days	28 days	56 days	90 days	180 days
OPSLWC– CTR–0%	18.66 (74.6%)	24.23 (96.8%)	25.02 (100%)	25.95 (103.7%)	25.99 (103.9%)	26.88 (107.4%)
OPSLWC– MPS–0.1%	23.09 (88.1%)	24.68 (94.1%)	26.22 (100%)	26.80 (102.2%)	27.00 (103%)	30.07 (114.7%)
OPSLWC– BPS–0.3%	20.45 (75.6%)	23.53 (87%)	27.04 (100%)	27.30 (101%)	27.70 (102.4%)	30.39 (112.4%)
OPSLWC– HYB–0.4%	21.05 (71.6%)	24.19 (82.3%)	29.38 (100%)	29.88 (101.7%)	29.92 (101.8%)	31.45 (107%)

The data in parentheses are percentage of 28-day compressive strength.

**Table 8 materials-14-02337-t008:** Residual compressive strength of all mixes at 28 days.

Mix Code	FLCS (MPa)	SLCS (MPa)	TLCS (MPa)
OPSLWC–CTR–0%	25.02	16.22 (64.8%)	10.24 (40.9%)
OPSLWC–MPS–0.1%	26.22	21.66 (82.6%)	19.5 (74.4%)
OPSLWC–BPS–0.3%	27.04	23.53 (87.0%)	20.21 (74.7%)
OPSLWC–HYB–0.4%	29.38	26.14 (89.0%)	24.47 (83.3%)

The data in parentheses are percentage of FLCS.

**Table 9 materials-14-02337-t009:** Residual splitting tensile strength of all mixes at 28 days.

Mix Code	FLSTS (MPa)	SLSTS (MPa)	TLSTS (MPa)
OPSLWC–CTR–0%	2.08	–	–
OPSLWC–MPS–0.1%	2.09	1.06 (50.7%)	0.61 (29.2%)
OPSLWC–BPS–0.3%	2.11	1.32 (62.6%)	1.02 (48.3%)
OPSLWC–HYB–0.4%	2.15	1.74 (80.9%)	1.43 (66.5%)

The data in parentheses are percentage of FLSTS.

## Data Availability

Data is contained within the article.

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
