# Peer review of "Mechanical and Thermal Properties of Synthetic Polypropylene Fiber–Reinforced Renewable Oil Palm Shell Lightweight Concrete"

_materials, 2021, doi:10.3390/ma14092337_

Round 1

Reviewer 1 Report

The manuscript presents very interesting results on lightweight concrete with oil palm shell and synthetic polypropylene fibers. I feel the manuscript missed a part to discuss the effects of oil palm shell on the results. Other comments are given:

In Figure 1, when did you add fibers?

Line 186: “ODD is calculated 186 with the weight of the specimens measured after being oven dried for 24 hours ”, can you specify the sample age and drying temperature?

Line 224: “This phenomenon could be due to the mixing method. The expedited reaction of SP and cement was expedited could result a more thorough dispersion of cement particles.”  Compressive strength increasing with age is normal. Can you elaborate on how this is linked to the mixing method? The effect of SP on cement hydration is generally expected in a few days. For the long term, can you explain why you think SP still affects hydration?

Line 234: “This test was conducted by further loading the specimens for 234 second time and third time at 28 days.” You may need to provide more information about these tests, such as time intervals between these tests.

Line: “Voids present in 353 the mix at the age of 7-days due to the cement was in partially hydrated stage [31]. The further 354 hydration of cement in the mix at 28-days produced calcium silica hydrate to fill up most of the 355 voids in the mix. ” Air voids may be partially filled with hydration products, but rarely “most of voids” being filled with hydration products, so this effect is not significant. The increase of thermal conductivity is simply due to the increase of solid phases, which have higher thermal conductivity than air and water.

Author Response

We are grateful to the Reviewers for many valuable and constructive comments; we do appreciate the time and efforts and without these important comments, this article would not have been improved to this extent.

Reviewer 2 Report

The paper presents the results of an experimental study on an ecological lightweight concrete in which part of the aggregate is replaced with an oil palm shell. The authors investigate the mechanical properties of such concrete to conclude whether it can be used in structural applications.

The subject of the study is of course very interesting because of the increasing demand for "green" materials and re-use of post-industrial waste, and OPS concrete has been proven by other authors as a promising material. However, the scientific and practical quality of the paper is disputable for two main reasons:

  1. The authors try to investigate the impact of too many variable parameters on both the mechanical and physical properties of the new lightweight concrete (replacement of aggregate, addition of dispersed reinforcement and with different types of fibre reinforcement) and it seems that they do not have full control on the research process.
  2. Having in mind the above, the range of experimental tests used for investigation of this behaviour is very poor.

Therefore, in my opinion, the authors cannot fully explain the behaviour of the concrete they observe and thus the obtained results should be treated with some reserve. 

Nevertheless, the authors have performed significant research work which has a potential for publication when the following remarks are addressed:

  1. I suggest that the authors limit this research only to the mechanical behaviour of this new concrete. There is still a lot to be done in this regard, while the study of physical (thermal) properties of this concrete is a completely different subject which also requires much broader analysis than just experimental investigation of thermal conductivity. E.g. heat capacity of concrete is also very important. When investigating thermal properties, the authors should first focus on deepened understanding of plain concrete, and only then move forward with analysis of fibre-reinforced concrete. The presented results show significant discrepancies and no trend, and the authors are not able to explain it.
  2. Regarding the analysis of mechanical properties of this lightweight concrete, I would first like the authors to explain their choices of concrete mix design:
    1. Please provide the sieve analysis curve of the applied aggregate (not only the maximum sizes of fine and coarse aggregate). Please explain according to what standard/specification the maximum size of 10 mm was chosen (normally, the max size of coarse aggregate in lightweight concrete is 8 mm).
    2. Please explain the very high cement content (515 kg is a lot for concrete and this will have an important impact on the strength, much higher than the aggregate used) and very low water-to-cement ratio (such concrete composition is similar to high-strength high-performance concretes). Also, with such a high level of Portland cement, it is highly disputable for me to call this concrete ecological.
  3. The study should begin with a comparison of OPS concrete with analogical "normal" lightweight concrete with standard aggregate. As stated above, such concretes are normally made with smaller amounts of cement, so no wonder that the authors obtained high strength by using very high cement contents. This strength should not be attributed to the use of oil palm shell. 
  4. The levels of the applied fibre reinforcement are very small and an attempt to find any trends in the behaviour of fibre-reinforced concrete with respect to the reinforcement degree with such a small difference in this degree is also disputable. Normally one should expect the analysis of fibre content of e.g. 0.5%, 1% and 1.5%. Please explain.
  5. In section 3.3 the authors conclude that their concrete meets the requirements for the structural lightweight concrete. Please provide the reference.
  6. The authors make an analysis of compressive strength development in time and draw very brave and detailed conclusions. In my opinion, the range of the performed research is too scarce to make such opinion based on it. Firstly, the concrete mix here is very atypical, mind especially very low w/c (limited access to water which may delay hardening process) - as stated already, please compare with the behaviour of high-strength concretes and their specific methods of mix design and methods of curing. I also advise checking if there is no chemical reaction between cement and oil palm shell which may influence the hydration process (e.g. thermogravimetry should be performed). Alternatively, if such investigations have been made on the OPS concrete, please give reference.  
  7. Further on the thermal conductivity study, please explain why there was no change of this property in time in plain concrete, while important development was observed when reinforcement was added? You refer to the effect of voids, but were there any experimental observations made? This would require microscopic inspection (in this case the sample must be destroyed) or tomography (this can be performed on whole sample).
  8. I lack crucial standard tests performed on fibre-reinforced concrete which characterise the impact of fibre reinforcement of mechanical properties, i.e. flexural tests and determination of toughness. 

Author Response

(The authors gave the same response as above.)

Reviewer 3 Report

Dear authors,
Concrete with the addition of palm oil shell is known, but this work is interesting and contributes to the analysis of such concrete. The quality of the work would be contributed by the addition in the conclusion which would refer to the possibilities of application of such concrete in the construction. For example, could it be used in the construction of load-bearing elements, or only for partition, non-load-bearing parts, decoration on facades, etc.?
Below you can find some formal errors that I noticed in the paper.

Please check eq (1) and the same Eq at the diagram in fig3 where you presenting the relationship between Slump Value (cm) and Vebe Time (s).  It should be the same, isn't it?

In line 66 instead of thr, it could be „the“

Best regards

Author Response

(The authors gave the same response as above.)

Reviewer 4 Report

The review of the paper entitled “Mechanical and Thermal Properties of Synthetic Polypropylene Fibre-Reinforced Renewable Oil Palm Shell Lightweight Concrete” has been done. In this paper the authors study the mechanical and thermal properties of the concrete prepared by oil palm shell (OPS) aggregates reinforced with polypropylene fibres. The paper is well-written with good readability for the general reader of the journal. Also, the methods are carefully done and the results can be considered for publication in Materials Journal with some minor revision. My suggestion would be to add a brief paragraph about the use of nanomaterials in concrete technology in the Introduction section especially in the field where supplementary materials, such as metakaolin, fly-ash, or natural pozzolans, are used instead of cement or usual aggregates. The reason for this is that although many studies have been carried out about the effect of replacing normal aggregates with OPS aggregates on properties of concrete but there is still lack of studies about the role of nanomaterials on such replacements. This can give a good perspective to the journal reader for the future works that can be done in this field. I highly recommend the addition of the following references to the article for this paragraph. I believe the following published papers should be added to the Introduction and References sections:

Hamada, H.M., Thomas, B.S., Tayeh, B., Yahaya, F.M., Muthusamy, K. and Yang, J., 2020. Use of oil palm shell as an aggregate in cement concrete: A review. Construction and Building Materials, 265, p. 120357.

Shakiba, M., Rahgozar, P., Elahi, A.R. and Rahgozar, R., 2018. Effect of activated pozzolan with Ca(OH)2 and nano-SiO2 on microstructure and hydration of high-volume natural pozzolan paste. Civ Eng. J, 4(10), pp. 2437-2449.

Zhang, P., Li, Q., Chen, Y., Shi, Y. and Ling, Y.F., 2019. Durability of steel fiber-reinforced concrete containing SiO2 nano-particles. Materials, 12(13), p. 2184.

Norhasri, M.M., Hamidah, M.S. and Fadzil, A.M., 2017. Applications of using nano material in concrete: A review. Construction and Building Materials, 133, pp. 91-97.

Author Response

Again, we are grateful to the Reviewers for many valuable and constructive comments; we do appreciate the time and efforts and without these important comments, this article would not have been improved to this extent.

Round 2

Reviewer 2 Report

Thank you for your quick response. Nevertheless, I must admit that I am not at all satisfied with the response to my remarks. The authors have introduced some minor comments to the methods and results, but the most important issues have not been addressed nor amended. The presented research program is incomplete and in some places improperly designed, also not in line with the current state of the art. I will not repeat all my comments, just to point the most crucial issues:

  1. A comparison should be made to analogical normal lightweight concrete (with standard aggregate). This should be a reference concrete.
  2. Mix design with 515 kg/m3 of cement in a potentially ecological concrete is unacceptable. Also, such a design does not prove that oil shell can serve as an effective aggregate replacement in lightweight concrete for structural use; this is achieved solely by the high amount of cement and low w/c ratio.
  3. Thermogravimetric tests and microscopic images are required to investigate the maturity development in such concrete. Otherwise, this strange age dependency is vaguely explained.
  4. It is better to omit the analysis of thermal properties with such little research on these properties. Otherwise, more extensive tests are required.
  5. If the authors want to investigate the effect of fibres, the state-of-the-art test is a 3-point bending with a notch/4-point bending and determination of toughness; only this value can tell us anything about the influence of fibres on mechanical properties. 

In the current form, I may recommend the paper to be presented as a conference paper (due to the preliminary character of the presented research) but I do not find it suitable for journal publication.

Author Response

(The authors gave the same response as above.)
